# Exhaled Hydrogen as a Marker of Intestinal Fermentation Is Associated with Diarrhea in Kidney Transplant Recipients

**DOI:** 10.3390/jcm10132854

**Published:** 2021-06-28

**Authors:** Fernanda Guedes Rodrigues, J. Casper Swarte, Rianne M. Douwes, Tim J. Knobbe, Camilo G. Sotomayor, Hans Blokzijl, Rinse K. Weersma, Ita P. Heilberg, Stephan J. L. Bakker, Martin H. de Borst

**Affiliations:** 1Department of Nephrology, University Medical Center Groningen, University of Groningen, 9700 RB Groningen, The Netherlands; j.c.swarte@umcg.nl (J.C.S.); r.m.douwes@umcg.nl (R.M.D.); t.j.knobbe@umcg.nl (T.J.K.); c.g.sotomayor.campos@umcg.nl (C.G.S.); s.j.l.bakker@umcg.nl (S.J.L.B.); m.h.de.borst@umcg.nl (M.H.d.B.); 2Nutrition Post Graduation Program, Universidade Federal de São Paulo, São Paulo 04023-062, Brazil; ita.heilberg@gmail.com; 3Department of Gastroenterology and Hepatology, University Medical Center Groningen, University of Groningen, 9700 RB Groningen, The Netherlands; H.blokzijl@umcg.nl (H.B.); r.k.weersma@umcg.nl (R.K.W.); 4Division of Nephrology, Universidade Federal de São Paulo, São Paulo 04023-062, Brazil

**Keywords:** kidney transplantation, hydrogen, diarrhea, small intestinal bacterial overgrowth

## Abstract

Background: Diarrhea is common among kidney transplant recipients (KTR). Exhaled hydrogen (H_2_) is a surrogate marker of small bowel dysbiosis, which may drive diarrhea. We studied the relationship between exhaled H_2_ and diarrhea in KTR, and explored potential clinical and dietary determinants. Methods: Clinical, laboratory, and dietary data were analyzed from 424 KTR participating in the TransplantLines Biobank and Cohort Study (NCT03272841). Fasting exhaled H_2_ concentration was measured using a model DP Quintron Gas Chromatograph. Diarrhea was defined as fast transit time (types 6 and 7 according to the Bristol Stool Form Scale, BSFS) of 3 or more episodes per day. We studied the association between exhaled H_2_ and diarrhea with multivariable logistic regression analysis, and explored potential determinants using linear regression. Results: KTR (55.4 ± 13.2 years, 60.8% male, mean eGFR 49.8 ± 19.1 mL/min/1.73 m^2^) had a median exhaled H_2_ of 11 (5.0–25.0) ppm. Signs of small intestinal bacterial overgrowth (exhaled H_2_ ≥ 20 ppm) were present in 31.6% of the KTR, and 33.0% had diarrhea. Exhaled H_2_ was associated with an increased risk of diarrhea (odds ratio 1.51, 95% confidence interval 1.07–2.14 per log_2_ ppm, *p* = 0.02). Polysaccharide intake was independently associated with higher H_2_ (std. β 0.24, *p* = 0.01), and a trend for an association with proton-pump inhibitor use was observed (std. β 0.16 *p* = 0.05). Conclusion: Higher exhaled H_2_ is associated with an increased risk of diarrhea in KTR. Our findings set the stage for further studies investigating the relationship between dietary factors, small bowel dysbiosis, and diarrhea after kidney transplantation.

## 1. Introduction

Kidney transplantation is the preferred treatment for end-stage kidney disease (ESKD) [1]. Given the advances in surgical techniques and immunosuppressive therapy in parallel with prophylaxis and treatment of infectious complications in the past decades, patient and graft short-term outcomes have considerably improved [2,3]. However, the quality of life of many outpatient kidney transplant recipients (KTR) is adversely affected by late complications, including diabetes, malignancies, risk of opportunistic infections due to maintenance immunosuppressive therapy, and gastrointestinal (GI) complaints [4,5]. GI complaints affect 30 to 40% of these patients, with chronic diarrhea impacting the quality of life of 20% of otherwise stable KTR during the first year after kidney transplantation [6].

It has recently been shown that KTR are commonly affected by an unbalanced gut microbiome, i.e., gut dysbiosis, characterized by a diminished microbial diversity [7,8]. Emerging evidence indicates that changes in gut microbiota following kidney transplantation may play a key role in the development of GI symptoms and diarrhea [7,9]. Small intestinal bacterial overgrowth is a form of gut dysbiosis characterized by an excessive number of coliform bacteria in the upper part of the small bowel, which has been implicated in driving GI complaints such as diarrhea, abdominal pain, and bloating [10]. In the presence of small bowel dysbiosis, the conversion of substrates into short-chain fatty acids (SCFA) is shifted to a higher production of intestinal gases such as hydrogen (H_2_), carbon dioxide (CO_2_), and methane (CH_4_) [11]. Exhaled hydrogen (H_2_) can be used as a non-invasive surrogate marker for small intestinal bacterial overgrowth [12]. Whether exhaled H_2_ is associated with the risk of diarrhea in KTR is currently unknown, and the role of post-kidney transplant medication regimens (e.g., maintenance immunosuppressive therapy and proton-pump inhibitors) and diet composition as potential determinants of exhaled H_2_ have not been investigated.

Therefore, in the current study we assessed exhaled H_2_ in a large KTR cohort, to study its relationship with diarrhea and to investigate its potential clinical and dietary determinants.

## 2. Materials and Methods

This is a cross-sectional study based on data from the TransplantLines Biobank and Cohort Study (ClinicalTrials.gov identifier: NCT03272841), conducted at the outpatient clinic of the University Medical Centre Groningen (Groningen, The Netherlands), which investigates all different types of solid organ transplant recipients [1]. A detailed description of the study design, inclusion and exclusion criteria has been described previously [1]. The study protocol has been approved by the Institutional Review Board (METc 2014/077) (METc UMCG), adheres to the local UMCG Biobank Regulations, and is in accordance with the WMA Declaration of Helsinki and the Declaration of Istanbul [2]. KTR with available breath test data were included in the present study and all participants signed an informed consent prior to their TransplantLines visit. A flowchart diagram is presented in Figure 1.

### 2.1. Clinical Data

Clinical data were collected according to a detailed protocol, as described elsewhere [1]. Patients were recruited between January 2017 and June 2019. Patients taking antihypertensive drugs were classified as having hypertension [3]. Diabetes mellitus was defined according to the guidelines of the American Diabetes Association [4]. Body mass index (BMI) was calculated as weight in kilograms divided by height in meters squared (kg/m^2^). Body surface area (BSA) was calculated using the formula of Du Bois and Du Bois [5]. Estimated glomerular filtration rate (eGFR) was calculated using the CKD Epidemiology Collaboration (CKD-EPI) creatinine equation [6]. Body composition was determined using a multifrequency bioelectrical impedance device (BIA, Quadscan 4000, Bodystat, Douglas, British Isles) at 5, 50, 100, and 200 Hz, which allows us to distinguish between lean mass and fat mass (expressed as body fat percentage) taking into account differences in volume status [7].

### 2.2. Breath H_2_ Measurement

Breath samples were collected in a 50 cc syringe with a hole of 6 mm at approximately 40 cc with a 3-way-stopcock. Patients were fasting (and therefore did not ingest any carbohydrates) for at least eight hours and were not allowed to smoke for at least one hour before the sample collection [12]. The study subject inhaled normally and exhaled maximally in this syringe with the stamper set at 50 cc and the 3-way stopcock open. After full expiration, the hole was immediately closed by the study participant, the stamper was set to 30 cc, and the 3-way stopcock was closed. Breath samples were analyzed within 12 h after sample collection using a model DP Quintron Gas Chromatograph (Quintron Instrument Company, Milwaukee, WI, USA). H_2_ results were automatically corrected for CO_2_ in order to reduce the chance of dilution by environmental air. A fasting basal (i.e., without the ingestion of test sugar) exhaled H_2_ concentration above 20 parts per million (ppm) was considered suggestively positive for small intestinal bacterial overgrowth (SIBO) as suggested elsewhere [8,9,10,11].

### 2.3. Stool Water Content Measurement

One day before the study visit, participants had collected a stool sample at home with a FaecesCatcher (TAG Hemi VOF, Zeijen, The Netherlands) [1]. The sample was collected in a tube and immediately frozen. At arrival, the sample was immediately stored at −80 °C (−112 °F) until further use. For analysis, samples were defrosted up to ~0 °C and homogenized. Then, a minimum of 1 g and preferably 5 g of every sample was put in a 15 mL tube for stool water content measurement. Prior to filling, two holes were pierced in the lid to allow water sublimation during freeze-drying. Subsequently, samples were freeze-dried for 48 h under 0.5 bar at −50 °C [13]. The samples were weighed before and after freeze-drying to calculate the dry weight as shown in the equation below [14].

Equation: Percentage of dry matter stool samples.
$$Dry\;matter\;\%=\frac{\left(Dry\;filled\;tube-empty\;tube\right)}{\left(Wet\;filled\;tube-empty\;tube\right)}\times 100\%$$

### 2.4. Diarrhea Classification

The stool form and consistency were graded using the Bristol Stool Form Scale (BSFS) [15]. The scale is structured from 1 to 7 according to form and consistency, from the hardest (type 1) to the most fluid kind (type 7). KTR classified as slow transit time (type-1 and -2 feces in the Bristol scale) and normal transit time (types 3, 4, and 5) were clustered as having no diarrhea, and those with fast transit time (types 6 and 7) with 3 or more episodes per day, as having diarrhea.

### 2.5. Dietary Assessment

Dietary intake was assessed using a validated self-administered food frequency questionnaire (FFQ) [16]. A trained researcher checked the FFQ for completeness on the day of the visit to the outpatient clinic. The FFQ inquired about consumption of 177 food items during the past month, taking seasonal variations into account, and included 7 fruit items and 18 vegetable items. Frequency was recorded in times per day, week, or month, and servings were expressed as natural units or household measures. The FFQ was linked to the Dutch Food Composition Table (NEVO) in order to calculate total energy intake and nutrients [17]. Adjustment for total energy intake according to the residual method was performed to calculate nutrients intake [18].

### 2.6. Statistical Analyses

Statistical analyses were performed using IBM SPSS version 23.0 (SPSS Inc., Chicago, IL, USA). In all analyses, *p* < 0.05 was considered significant. Variable distribution was evaluated by Kolmogorov–Smirnov test. Categorical variables are presented as n (%), normally distributed variables as mean ± standard deviation (SD), and non-normally distributed variables as median (interquartile range). We divided patients into three groups according to exhaled hydrogen. The highest group was defined as >20 ppm (suggestive for SIBO [8,9,10,11]). We divided the remaining patients (exhaled H2 < 20 ppm) into two groups using the rank tool in SPSS software. Comparison of categorical variables was performed using a Chi-square test. Differences in groups of exhaled H_2_ were tested through analysis of variance (ANOVA) with Bonferroni post hoc tests for normally distributed variables and the Kruskal–Wallis test for non-normal distribution. Possible determinants of exhaled H_2_ were studied using univariable linear regression. Since we aimed to explore the potential relevance of any clinical or nutritional factor as potential determinant of exhaled H_2_, we tested all available variables in individual univariable regression analysis. Subsequently, all variables with a *p* < 0.05 were included in a multivariable linear regression model to identify independent determinants of exhaled H_2_ production. Residuals were checked for normality and variables were natural log-transformed when appropriate. Multivariable logistic regression was performed to determine the potential relationship between exhaled H_2_ and diarrhea. Variables of known clinical importance for diarrhea in KTR, such as age, sex, eGFR, transplant vintage, immunosuppressive use [19,20], and H_2_ determinants, such as use of PPI and polysaccharides intake, were used in the model.

## 3. Results

### 3.1. Clinical Parameters

A total of 424 KTR (55.4 ± 13.2 years, 258 (60.8%) male) were included in the study. The median transplant vintage was 1.8 (1.0–7.2) years, and 16.3% had a history of allograft rejection. With respect to comorbidities, 19.3% and 73.3% of patients had diabetes and hypertension, respectively. The mean eGFR was 49.8 ± 19.1 mL/min/1.73 m^2^. According to the BSFS, 33% of KTR had diarrhea. Further clinical and laboratory characteristics are shown in Table 1. KTR were divided into three groups according to exhaled H_2_ concentration (G1, 1.0–6.9 ppm, n = 151; G2, 7.0–19.9 ppm, n = 139; G3, ≥20.0 ppm, n = 134). One-hundred thirty-four (134) out of 424 patients (31.6%) were considered positive for small intestinal bacterial overgrowth (H_2_ ≥ 20 ppm). KTR in the highest H_2_ group had lower BMI (26.8 ± 3.9 kg/m^2^ vs. 28.2 ± 5.6 kg/m^2^, *p* = 0.02) and body fat percentage (29.3 ± 9.0% vs. 32.3 ± 10.3%, *p* = 0.02) when compared to the lowest H_2_ group. Patients in the highest H_2_ group also had lower waist circumference when compared to both the lowest and intermediate group (96.9 ± 12.1 cm vs. 100.8 ± 13.6 cm and 100.8 ± 13.4 cm, *p* = 0.03).

The immunosuppressive regimen consisted mostly of triple therapy including prednisolone (in 93.4% of all patients) and mycophenolate mofetil (MMF) (73.3%) in combination with the calcineurin inhibitor tacrolimus (78.5%) or cyclosporine (9.0%). Alternatively, regimens could contain azathioprine (5.7%) or everolimus (2.4%). When comparing the groups, we did not find statistical differences regarding immunosuppressive regimens. When comparing immunosuppressive trough blood concentrations, the highest H_2_ group had significantly higher levels of tacrolimus when compared to the lowest H_2_ group, and a trend when compared to the intermediate group (6.8 ± 2.4 ng/mL vs. 6.1 ± 2.2 ng/mL, *p* = 0.07). KTR in the intermediate (n = 105, 78.4%) and highest (n = 105, 75.5%) H_2_ groups more commonly used proton-pump inhibitors when compared to the lowest group (n = 96, 63.5%), *p* = 0.01, while use of statins and antihypertensive drugs were similar among the groups. Nutritional data (Appendix A) showed no statistical differences regarding energy intake, macronutrients (protein, carbohydrates, and lipids), fiber, and micronutrients, including calcium, iron, zinc, and vitamins D, C, and E, among the three groups based on exhaled H_2_. Patients in the highest H_2_ group had lower intake of mono/disaccharides (113.6 ± 47.7 g vs. 126.2 ± 45.4 g and 124.9 ± 35.5 g, *p* = 0.02) and higher intake of polysaccharides (161.0 ± 36.3 g vs. 143.2 ± 38.7 g and 144.3 ± 28.3 g, *p* = 0.004) when compared to the lowest and intermediate groups, respectively.

### 3.2. Determinants of Exhaled H_2_

We subsequently performed linear regression to investigate possible clinical, laboratory and dietary factors determinants of exhaled H_2_. Associations were explored for all factors shown in Table 1; (borderline) significant results are presented in Table 2. Upon univariable analysis, we observed inverse associations between H_2_ and mono/disaccharides (std. β −0.27, *p* = 0.01) and vitamin C intake (std. β −0.17, *p* = 0.03), as well as a positive association with polysaccharide intake (std. β 0.49, *p* < 0.001). Other possible influencing factors, such as waist circumference (WC), total cholesterol, low-density lipoprotein (LDL)-cholesterol, tacrolimus trough blood, and use of proton-pump inhibitors were also significant in univariable analyses. In multivariable linear regression analyses, only polysaccharide intake remained independently associated with exhaled H_2_ (std. β 0.24, *p* = 0.01), while a trend for an association with proton-pump inhibitor use was observed (std. β 0.16, *p* = 0.05).

### 3.3. H_2_ and Diarrhea

Table 3 summarizes the results of logistic regression, which revealed that exhaled H_2_ and the use of MMF were significantly associated with diarrhea according to BSFS (OR = 1.51, 95% CI 1.07–2.14 per log_2_ ppm, *p* = 0.02, and OR = 4.71 95% CI 1.24–17.77, *p* = 0.02, respectively), while adjusting for potential confounders including age, sex, eGFR, transplant vintage, tacrolimus use, and polysaccharides intake. Although individuals with higher stool water content also had a higher number of evacuation episodes (r = 0.45, *p* = 0.01), the number of evacuation episodes was not associated with H_2_ (r = −0.02, *p* = 0.75).

Feces samples of 75 KTR were available for the analysis of water content. Patients in the highest H_2_ group displayed a trend towards higher percentage of water stool content when compared to the lowest group (77.7 ± 5.5% vs. 73.8 ± 6.0%, *p* = 0.08). Linear regression analysis disclosed a trend for an association between exhaled H_2_ and stool water content (std. β 0.22, *p* = 0.06; data not shown in tables).

## 4. Discussion

Diarrhea is a common complication after kidney transplantation [21,22], with its etiology still under debate. In the current study, 33.0% of KTR had diarrhea according to the BSFS questionnaire, and 31.6% presented exhaled H_2_ higher than 20 ppm. In multivariable-adjusted analyses, exhaled H_2_ was associated with an increased risk of diarrhea in KTR. We identified polysaccharide intake as an independent dietary determinant of exhaled H_2_. The present data suggest a relationship between small bowel dysbiosis, diarrhea, and diet after kidney transplantation.

As a frequent GI symptom after kidney transplantation, diarrhea may be related to infections, antibiotics, or immunosuppressive drugs [20]. In the present study, 33.0% of KTR had diarrhea, in line with the 35.0% described by Lee et al. [23], but lower than the 53.0% presented by Ekberg et al. [21]. This variable prevalence could be attributed to differences in diarrhea classification, immunosuppressive regimens, and sample size. Nevertheless, the prevalence of diarrhea in KTR far exceeded the prevalence observed in the general population (3.5 to 12.0%) [24,25].

We observed independent associations between both exhaled H_2_ and MMF use with the presence of diarrhea. MMF use has consistently been implicated in posttransplant diarrhea [26]. Our finding regarding the relationship between exhaled H_2_ and diarrhea is in line with previous data outside the transplant population [27]. In irritable bowel syndrome (IBS) patients with small intestinal bacterial overgrowth (SIBO), a type of gut microbiome dysbiosis, was present more often with diarrhea, higher stool frequency, and looser stool forms. Moreover, these symptoms were associated with higher bacterial count in upper gut aspirate and basal exhaled H_2_ in both fasting state and following ingestion of a substrate [28].

Although our study did not reveal potential mechanisms underlying the association between exhaled H_2_ and diarrhea, we hypothesize that lipopolysaccharide (LPS) present in Gram-negative bacteria, the most common microorganisms related to SIBO, may promote local inflammation, causing mucosal lesions and increasing intestinal permeability, disabsorption syndrome, and increased nutrient fermentation [29]. Because the production of H_2_ in humans only occurs through microbial anaerobic fermentation of unabsorbed carbohydrates [30], higher exhaled H_2_ is generally considered a marker for alterations in small bowel microbiota composition. Since higher levels of exhaled H_2_ can be caused either by slow transit or by bacterial overgrowth with a delayed return of exhaled H_2_ to baseline levels, Romagnuolo et al. [10] states that fasting exhaled H_2_ levels ≥20 ppm can be representative of SIBO. In agreement, Corazza et al. [31] have demonstrated that, in bacterial overgrowth patients, fasting exhaled H_2_ values were significantly higher than in healthy volunteers (14.7 ± 14.0 ppm vs. 5.8 ± 3.1 ppm, *p* < 0.001) [31]. Perman et al. [32] observed a fasting exhaled H_2_ of 2.0 ± 2.5 ppm after a dinner meal in healthy subjects, with no value exceeding 11.0 ppm, whereas exhaled H_2_ after an identical meal in patients with bacterial overgrowth exceeded 48 ppm [32]. These studies support that elevated values of fasting H_2_ can be considered suggestive of SIBO, potentially connecting higher exhaled H_2_ with the observed increased risk of diarrhea.

In addition, patients with higher exhaled H_2_ tended to have higher stool water content and the percentage of stool water content was positively associated with both exhaled H_2_ and the number of evacuation episodes. An experimental study has disclosed that greater gastro-intestinal H_2_ content shortened colonic transit time by 47% in the proximal colon, and by 10% in the distal colon [33]. These data suggest that there might be an association between an accelerated intestinal transit causing diarrhea and the H_2_ production.

A recent meta-analysis suggested that the use of proton-pump inhibitors (PPI) can moderately increase the risk of SIBO [34]. Since gastric acid is an important barrier that prevents bacterial colonization of the stomach and small intestine, PPI therapy may promote small intestinal microbiota growth, through chronic acid suppression and subsequent hypochlorhydria [35]. In the current study, we observed a significantly higher use of PPI among individuals in the highest group of exhaled H_2_, with a trend for an association with H_2_ in multiple linear regression, which is in line with the notion that exhaled H_2_ is associated with small intestinal microbiota overgrowth.

A complex interplay exists between diet, GI transit, and gut microbiota [36]. In the present study, we identified polysaccharide intake as an independent determinant of exhaled H_2._ Polysaccharides are complex carbohydrates that can be divided into starch and non-starch [37]. Since there was no difference in fiber intake between the H_2_ groups, we assume higher starch intake as the main cause of higher polysaccharide intake. Some starch, known as resistant starch (RS), escapes digestion in the small intestine and, upon reaching the large intestine, acts similarly to dietary fiber, fermenting and incorporating water [37]. Our findings are at least partly in line with a previous study suggesting that SIBO resulted from differences in fiber intake [38]. In the current cohort, the suggestive presence of SIBO could further promote starch malabsorption, which in turn could be a conceivable explanation for the higher water content in stool due to osmotic activity, subsequently causing diarrhea [39,40]. At the same time, there is limited evidence that SIBO is the primary driver of GI symptoms or that it is influenced by dietary factors.

RS and other types of starch that escape digestion in the small intestine may be quantitatively more important as substrates of fermentation than non-starch polysaccharides in the colon [41]. While the human genome does not encode adequate gastrointestinal enzymes that metabolize some polysaccharides, RS undergoes fermentation by members of the gut microbiota, resulting in the production of SCFAs, mainly butyrate [42,43]. Recently, our group has been able to show that the gut microbiome of KTR contained less butyrate-producing bacteria, more Proteobacteria, and fewer Actinobacteria [44]. Starch-utilizing bacteria in the gut include many members of the *Bacteroides* and various *Firmicutes* [45]. A study in KTR demonstrated that the phylum Bacteroidetes and its derivative Bacteroides, as well as *Ruminococcus*, *Coprococcus*, and *Dorea*, were significantly reduced in fecal specimens from patients with diarrhea [23,45]. These observations suggest that changes in starch-degrading bacteria in KTR could decrease carbohydrate fermentation in colonic lumen, promoting reduced nutrient absorption and watery stool.

Another important fact to be highlighted is that the elimination of the H_2_ produced by bacterial fermentation depends significantly on methanogenic and sulfate-reducing bacteria that convert H_2_ to methane and hydrogen sulfide [46]. We recently demonstrated that colonic presence of the methanogen *Methanobrevibacter smithii*, which plays an important role upon removing the end-product H_2_ from bacterial fermentation, was reduced in KTR [47]. These findings may indicate that diminished abundance of methanogens after kidney transplantation could lead to less H_2_ metabolization, also contributing to a rise in the gas levels in the breath test.

To the best of our knowledge, this is the first study to evaluate the relationship between exhaled H_2_ and diarrhea in a large-scale cohort of KTR. The breath test we used is widely available, safe, inexpensive, and noninvasive, advantages that make it ideal for daily clinical practice employment. Limitations of the present study include the cross-sectional nature, precluding conclusions on causality, the single H_2_ measurement, which does not necessarily reflect the entire post-transplant period, and the lack of information regarding the course and duration of diarrhea (acute or chronic). Since our study population consisted almost entirely of Caucasians, our results cannot be extrapolated to different populations. The purpose of performing the breath test was to investigate exhaled H_2_ in KTR under fasting basal conditions (i.e., without using substrates like glucose, lactulose, lactose, or fructose). Although some investigators suggest that the use of fasting hydrogen is insufficient for SIBO diagnosis [48], fasting hydrogen overproduction has been consistently found with bacterial overgrowth in several studies [8,10,32,33]. Finally, no direct tests such as duodenal aspirate cultures have been performed to detect changes in the gut microbiota colonization of small bowel.

In conclusion, a fasting exhaled H_2_ higher than 20 ppm was present in 31.6% of KTR, which was associated with increased risk of diarrhea. Polysaccharide intake was an independent determinant of exhaled H_2_. The present results suggest that diarrhea in KTR may reflect an altered small bowel gut microbial composition, at least partly under dietary control. These data encourage future studies to validate our findings; to further investigate the associations between the diet, small bowel dysbiosis, and post-kidney transplant diarrhea; and to explore whether lowering polysaccharide intake or correction of SIBO may reduce diarrhea in KTR. Characterizing small intestinal bacterial overgrowth in post-transplant patients with GI symptoms could support more focused antibacterial or dietary therapeutic approaches.

## Figures and Tables

**Figure 1 jcm-10-02854-f001:**
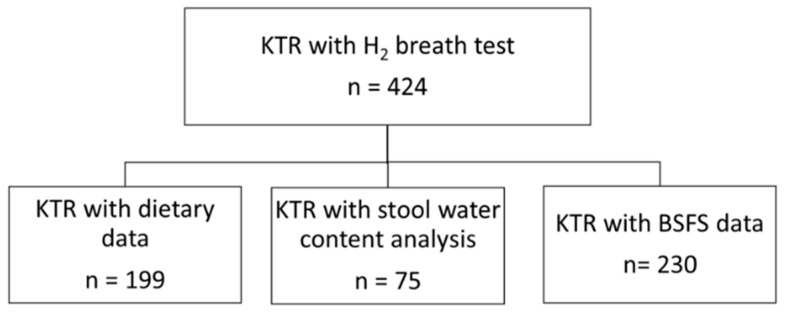
Flowchart diagram. Abbreviations: H_2_, hydrogen; KTR, kidney transplant recipients; BSFS, Bristol Stool Form Scale.

**Table 1 jcm-10-02854-t001:** Baseline characteristics of KTR according to groups of exhaled H_2_.

Baseline Characteristics	Total (n = 424)	Exhaled H_2_, per Group		
		1.0–6.9 ppm (n = 151)	7.0–19.9 ppm (n = 139)	≥20.0 ppm (n = 134)
**Fermentation Parameter**				
H_2_, ppm	11 (5.0–25.0)	4.0 (2.0–5.0)	11.0 (8.0–15.0) ^b^	33.5 (26.0–49.0) ^ab^
**Demographics**				
Age, years	55.4 ± 13.2	55.4 ± 13.7	57.2 ± 12.1	53.7 ± 13.5
Sex (male), *n* (%)	258 (60.8)	86 (57.0)	84 (62.7)	88 (63.3)
Transplant vintage, years	1.8 (1.0–7.1)	2.0 (1.0–8.1)	1.0 (0.6–5.0)	1.1 (0.8–7.8)
History of allograft rejection, *n* (*%*)	69 (16.3)	27 (17.8)	19 (14.2)	23 (16.5)
**Body Composition**				
Body mass index, kg/m^2^	27.7 ± 4.8	28.2 ± 5.6	28.1 ± 4.6	26.8 ± 3.9 ^a^
Waist circumference, cm	99.5 ± 13.2	100.8 ± 13.6	100.8 ± 13.4	96.9 ± 12.1 ^ab^
Body fat percentage, %	31.1 ± 9.8	32.3 ± 10.3	31.6 ± 9.8	29.3 ± 9.0 ^a^
**Immunosuppressive Drug Use**				
MMF, *n* (*%*)	311 (73.3)	113 (74.8)	100 (74.6)	98 (70.5)
Tacrolimus, *n* (*%*)	333 (78.5)	111 (73.5)	112 (83.5)	110 (79.1)
Cyclosporine, *n* (*%*)	38 (9.0)	17 (11.3)	12 (8.9)	9 (6.5)
Everolimus, *n* (*%*)	10 (2.4)	5 (3.3)	3 (2.2)	2 (1.4)
Prednisolone, *n* (*%*)	396 (93.4)	141 (93.4)	124 (92.5)	131 (94.2)
Azathioprine, *n* (*%*)	24 (5.7)	8 (5.3)	6 (4.5)	10 (7.2)
**Immunosuppressive Drug Trough Levels**				
MMF, ug/L	2.3 ± 1.6	2.2 ± 1.7	2.2 ±1.4	2.5 ± 1.8
Tacrolimus, ug/L	6.3 ± 2.5	5.9 ± 2.3	6.1 ± 2.2	6.8 ± 2.4 ^a^
**Lifestyle**				
Current smoker, *n* (%)	13 (3.1)	7 (4.6)	4 (3.0)	2 (1.4)
Alcohol consumption, g/day	1.5 (0.0–7.9)	1.7 (0.0–7.3)	1.0 (0.2–9.2)	1.6 (0.0–7.6)
**Glucose Homeostasis**				
Diabetes mellitus, *n* (%)	82 (19.3)	28 (18.5)	32 (23.0)	22 (16.4)
Plasma glucose, mmol/L	6.1 ± 1.7	6.2 ± 1.6	6.1 ± 1.8	6.0 ± 1.8
HbA1c, %	6.0 ± 1.7	6.0 ± 0.9	6.0 ± 0.8	5.9 ± 0.9
**Lipids**				
Total cholesterol, mmol/L	4.6 ± 1.0	4.7 ± 1.0	4.7 ± 1.0	4.4 ± 1.1
HDL-cholesterol, mmol/L	1.3 ± 0.4	1.4 ± 0.4	1.4 ± 0.4	1.3 ± 0.4
LDL-cholesterol, mmol/L	2.9 ± 0.9	2.9 ± 0.9	3.0 ± 0.9	2.7 ± 0.9
Triglycerides, mmol/L	1.8 ± 0.8	1.9 ± 0.9	1.9 ± 0.8	1.8 ± 0.9
**Cardiovascular**				
SBP, mmHg	137.4 ± 16.9	137.9 ± 16.2	137.2 ± 16.6	137.1 ± 16.6
DBP, mmHg	79.9 ± 11.1	79.8 ± 10.7	80.2 ± 11.1	79.6 ± 11.4
**Kidney Function**				
eGFR, mL/min/1.73 m^2^	49.8 ± 19.1	49.3 ± 18.9	49.5 ± 18.2	50.4 ± 5.9
Creatinine, µmol/L	155.3 ± 124.3	149.9 ± 92.2	152.4 ± 110.9	163.9 ± 161.6
Urinary protein excretion, g/24 h	0.1 ± 0.2	0.2 ± 0.3	0.1 ± 0.1	0.1 ± 0.2
**Medication**				
Proton pump inhibitors, *n* (%)	306 (72.2)	96 (63.5)	105 (78.4) ^a^	105 (75.5) ^a^
Statins, *n* (%)	220 (51.9)	80 (53.0)	72 (53.7)	68 (48.9)
Antihypertensive, *n* (%)	311 (73.3)	106 (70.2)	105 (78.4)	100 (71.9)
Diarrhea according to BSFS, *n* (%) *	76 (33.0)	24 (27.9)	25 (36.2)	27 (36.0)
Evacuation episodes, n/day *	2.1 ± 1.3	2.2 ± 1.3	2.2 ± 1.3	2.1 ± 1.3
Stool water content, % **	75.4 ± 6.3	73.8 ± 6.0	75.8 ± 7.0	77.7 ± 5.5 ^c^

^a^*p* < 0.05 vs. G1 ^b^
*p* < 0.05 vs. G2 ^c^
*p* = 0.08 vs. G1; * *n* = 230; ** *n* = 75. Data are presented as mean ± standard deviation (SD), median with interquartile ranges (IQRs), or number with percentages (%). Abbreviations: H_2_, hydrogen; MMF, mycophenolate mofetil; HbA1c, hemoglobin A1c; HDL, high density lipoprotein; LDL, Low density lipoprotein; SBP, systolic blood pressure; DBP, diastolic blood pressure; eGFR, estimated glomerular filtration rate; BSFS, Bristol Stool Form Scale.

**Table 2 jcm-10-02854-t002:** Potential determinants of Log_2_ exhaled H_2_.

Potential Determinants	Univariate		Multivariate *	
	Std. β	*p*	Std. β	*p*
Polysaccharides intake, g	0.266	<0.001	0.243	0.01
Proton pump inhibitor use	0.160	<0.01	0.164	0.05
Mono and disaccharides intake, g	−0.188	0.01		
Tacrolimus trough levels, ug/L	0.133	0.02		
Vitamin C intake, mg	−0.162	0.02		
Total cholesterol, mmol/L	−0.106	0.03		
LDL-cholesterol, mmol/L	−0.101	0.04		
Waist circumference, cm	−0.098	0.05		

*n* = 196. Linear regression analysis with exhaled H_2_ as dependent variable. * Run backwards. Std. β, standardized beta; LDL, low-density lipoprotein.

**Table 3 jcm-10-02854-t003:** Multivariable association between exhaled H_2_ and diarrhea.

Variable	OR (95% CI)	*p*
Log_2_ exhaled H_2_, ppm	1.51 (1.07–2.14)	0.02
Sex (male)	1.10 (0.41–2.99)	0.85
eGFR, mL/min/1.73 m^2^	0.95 (0.92–0.99)	0.01
Transplant vintage, years	0.99 (0.99–1.01)	0.08
MMF use	4.71 (1.24–17.77)	0.02
Tacrolimus use	0.25 (0.05–1.22)	0.09
PPI, use	1.09 (0.37–3.30)	0.86
Polysaccharides intake, g	0.99 (0.98–1.01)	0.12

*n* = 196. Multivariable-adjusted binary logistic regression analysis. Abbreviations: OR, Odds ratio; CI, confidence interval; H_2_, hydrogen, eGFR, estimated glomerular filtration rate; MMF, mycophenolate mofetil; PPI, proton-pump inhibitors.

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
