# Peer review of "Exhaled Hydrogen as a Marker of Intestinal Fermentation Is Associated with Diarrhea in Kidney Transplant Recipients"

_jcm, 2021, doi:10.3390/jcm10132854_

Round 1
Reviewer 1 Report
In this manuscript, Rodrigues et al. evaluate exhaled H2 in 424 kidney transplant recipients and evaluate its association with diet and post-transplant diarrhea. Among the 424 kidney transplant recipients, 199 had dietary data, 230 had Bristol stool scale, and 75 had stool water content analysis. Using linear regression, they find that kidney transplant recipients with the highest H2 group had lower BMI and body fat percentage and had more commonly prescribed PPI. They further find an association between polysaccharide intake and exhaled H2 in multivariable linear regression analyses. They then report that exhaled H2 and the use of MMF was associated with post-transplant diarrhea.
Overall, this is a very well written manuscript that provides a potential novel explanation for post-transplant diarrhea using exhaled H2 as a surrogate marker for small bowel intestinal bacterial overgrowth. To the best of my knowledge, this is the first study to evaluate exhaled H2 in kidney transplant recipients. With respect to post-transplant diarrhea, the etiology can be multifactorial: antibiotics, infectious, and MMF related. This provides a potentially new reason for post-transplant diarrhea. Furthermore, the authors link exhaled H2 to polysaccharide intake and also to increased stool water content, supporting the possibility that exhaled H2 is associated with small bowel intestinal bacterial overgrowth. I think that this study is timely and important for the transplant community.
I have a few suggestions that may help to improve interpretation of the data:
1) The authors link MMF and exhaled H2 to diarrhea. I wonder if there is any additive effect of MMF and H2 together. Can the authors compare the rates of diarrhea among these groups?
- a) MMF and >20 ppm H2
- b) MMF and <20 ppm H2
- c) no MMF and >20 ppm H2
- d) no MMF and <20 ppm H2
2) For section 3.2 determinants of exhaled H2, I’m not sure how easy the interpretation of the linear regression is for determinants of exhaled H2 ppm and how linear it truly is. For example, the amount of vitamin C intake is linearly associated with exhaled H2 ppm. I think it could be useful instead or a supplementary table to do a logistic regression with the variables with the dependent variable, the surrogate of small intestinal bacterial overgrowth (i.e. > 20 H2 ppm) so the logistic regression is predicting the surrogate of small intestinal bacterial growth and with this, the authors could do a multivariable logistic regression with significant variables.
3) Diarrhea was defined using the BSFS but do the authors have more information about the course of the diarrhea in the patients? Was higher exhaled H2 associated with prolonged diarrhea? If the authors have more data on this, it would be interesting but understandably this may not be available.
Minor points:
4) in each of the tables, it would be helpful to list the number of patients analyzed in the legends (the number seems to be different for each table based on Figure 1)
5) For Table 1, what is the rationale for 3 groups? I understand the cut off for >20 ppm but why the cut off of 1 – 6.9 and 7 – 19.9?
Author Response
Reviewer 1
1) The authors link MMF and exhaled H2 to diarrhea. I wonder if there is any additive effect of MMF and H2 together. Can the authors compare the rates of diarrhea among these groups?
a) MMF and >20 ppm H2
- b) MMF and <20 ppm H2
- c) no MMF and >20 ppm H2
- d) no MMF and <20 ppm H2
Response: We thank the reviewer for this suggestion. We have performed this analysis and we could not find a statistical difference between the patients with and without diarrhea clustered in the abovementioned groups.
|
|
Presence of Diarrhea |
|
|
|
|
NO |
YES |
P value (chi-square) |
|
MMF and >20 ppm H2 |
37 (64%) |
21 (36%) |
0.63 |
|
MMF and <20 ppm H2 |
80 (68%) |
37 (32%) |
|
|
no MMF and >20 ppm H2 |
13 (62%) |
8 (38%) |
|
|
no MMF and <20 ppm H2 |
24 (71%) |
10 (29%) |
|
The table provided above does not confirm a potential interaction between MMF use and exhaled hydrogen, although the numbers are small, especially in the groups of patients not on MMF. Therefore, we decided not to add these data to the manuscript.
2) For section 3.2 determinants of exhaled H2, I’m not sure how easy the interpretation of the linear regression is for determinants of exhaled H2 ppm and how linear it truly is. For example, the amount of vitamin C intake is linearly associated with exhaled H2 ppm. I think it could be useful instead or a supplementary table to do a logistic regression with the variables with the dependent variable, the surrogate of small intestinal bacterial overgrowth (i.e. > 20 H2 ppm) so the logistic regression is predicting the surrogate of small intestinal bacterial growth and with this, the authors could do a multivariable logistic regression with significant variables.
Response: We thank the reviewer for this very important comment. We have performed the binary logistic regression as suggested by the reviewer and introduced a supplementary table (Table S2) with the results. The results presented in this new table confirm that polysaccharides intake is the main determinant of exhaled hydrogen in kidney transplant recipients.
3) Diarrhea was defined using the BSFS but do the authors have more information about the course of the diarrhea in the patients? Was higher exhaled H2 associated with prolonged diarrhea? If the authors have more data on this, it would be interesting but understandably this may not be available.
Response: The reviewer is right to point out this limitation. Unfortunately, as already anticipated by the reviewer, we do not have data available to evaluate the course of diarrhea. This limitation was added to line 328-329.
Minor points:
4) in each of the tables, it would be helpful to list the number of patients analyzed in the legends (the number seems to be different for each table based on Figure 1)
Response: We thank the reviewer for this suggestion. This information was added to the legends of the tables.
5) For Table 1, what is the rationale for 3 groups? I understand the cut off for >20 ppm but why the cut off of 1 – 6.9 and 7 – 19.9?
Response: We thank the reviewer for this very important comment. We divided patients into three groups according to exhaled hydrogen using the rank tool in SPSS software. This tool uses percentiles to divide the remaining patients (exhaled H2 <20 ppm) into two groups. This was clarified in the Methods section.

Reviewer 2 Report
There is no mention of the duration of diarrhea to classify it as acute or chronic. Work-up of diarrhea such as tests for stool microscopy, culture and electrolytes, and findings on colonoscopy/biopsy etc. to exclude other etiologies is not provided. It is not mentioned as to whether there was a standardized protocol to guide patients before testing for breath hydrogen regarding the total duration of fasting and last intake of carbohydrates. It would be interesting to know in future studies if treatment of presumed small intestinal bacterial overgrowth improved diarrhea and/or reduced measured breath hydrogen levels.
Author Response
Reviewer 2
- There is no mention of the duration of diarrhea to classify it as acute or chronic. Work-up of diarrhea such as tests for stool microscopy, culture and electrolytes, and findings on colonoscopy/biopsy etc. to exclude other etiologies is not provided.
Response: We agree with the reviewer about this limitation of the present manuscript. It was added to the limitation paragraph, line 328-329.
- It is not mentioned as to whether there was a standardized protocol to guide patients before testing for breath hydrogen regarding the total duration of fasting and last intake of carbohydrates.
Response: We thank the reviewer for this comment. Patients were fasting for at least eight hours and were not allowed to smoke for at least one hour before the sample collection. In the present study, the breath test was not conducted after a carbohydrate intake (i.e. glucose, lactulose, etc). Therefore, also the last carbohydrate intake was at least 8 hours before sampling. This information was added to the manuscript (lines 93-95).
- It would be interesting to know in future studies if treatment of presumed small intestinal bacterial overgrowth improved diarrhea and/or reduced measured breath hydrogen levels.
Response: We thank the reviewer for the suggestion, which we now also mention in the revised manuscript (line 344).
